# Association between Lifestyle Modification and All-Cause, Cardiovascular, and Premature Mortality in Individuals with Non-Alcoholic Fatty Liver Disease

**DOI:** 10.3390/nu16132063

**Published:** 2024-06-28

**Authors:** Yanqiu Huang, Jinfan Xu, Yang Yang, Tingya Wan, Hui Wang, Xiaoguang Li

**Affiliations:** State Key Laboratory of Systems Medicine for Cancer, Center for Single-Cell Omics, School of Public Health, Shanghai Jiao Tong University School of Medicine, Shanghai 200025, China; huangyanqiu@sjtu.edu.cn (Y.H.); xujinfan2022@163.com (J.X.); yyang93@shsmu.edu.cn (Y.Y.); tinya2000@163.com (T.W.)

**Keywords:** healthy lifestyle, NAFLD, all-cause mortality, cardiovascular disease, psychological status

## Abstract

Background: This study is designed to explore the correlation between multiple healthy lifestyles within the framework of “lifestyle medicine”, and the mortality risk of nonalcoholic fatty liver disease (NAFLD). Methods: The National Health and Nutrition Examination Survey (NHANES) database was employed. The analysis consisted of 5542 participants with baseline NAFLD and 5542 matched non-NAFLD participants from the database. Lifestyle information, including five low risk factors advocated by lifestyle medicine (healthy diet, vigorous physical activity, healthy sleep duration, avoiding smoking, and maintaining a non-depressed psychological status), was collected through a baseline questionnaire. Cox proportional hazards regression models and Kaplan–Meier survival curve were used to evaluate risk of mortality. In addition, subgroups were analyzed according to gender, age, body mass index and waist circumference. Results: In total, 502 deaths (*n* = 181 deaths from cardiovascular disease (CVD)) were recorded among NAFLD participants after the median follow up duration of 6.5 years. In the multivariate-adjusted model, compared to participants with an unfavorable lifestyle (scoring 0–1), NAFLD participants with a favorable lifestyle (scoring 4–5) experienced a 56% reduction in all-cause mortality and a 66% reduction in CVD mortality. Maintaining an undepressed psychological state and adhering to vigorous exercise significantly reduced CVD mortality risk in NAFLD participants (HR, 0.64 [95% CI, 0.43–0.95]; HR, 0.54 [95% CI, 0.33–0.88]) while maintaining healthy sleep reduced premature mortality due to CVD by 31%. Conclusions: Healthy lifestyle, characterized by maintaining an undepressed mental state and healthy sleep, significantly mitigates the risk of all-cause, CVD, and premature mortality risk among NAFLD patients, with a particularly pronounced effect observed in female and obese subpopulations.

## 1. Introduction

Non-alcoholic fatty liver disease (NAFLD) is a chronic liver disorder characterized by hepatic steatosis that is not attributable to alcohol abuse or viral or autoimmune hepatitis. It is the predominant form of chronic liver disease. With the capacity to evolve to end-stage liver disease, including liver cancer, NAFLD has emerged as a novel risk factor for mortality of all-cause and cardiovascular disease (CVD) [1,2]. With the development of the economy and changes in people’s dietary structure and living habits, the incidence and prevalence of NAFLD is rapidly expanding globally, seriously jeopardizing human health. In the United States, the number of liver disease-related deaths in the NAFLD-endemic population is projected to be 78,300 in 2030, a 178% increase from 2015, with similar increases in the occurrence of decompensated cirrhosis and HCC [3].

The management of NAFLD is like that of other metabolic disorders in that lifestyle changes are the cornerstone and core of NAFLD management [4]. A variety of dietary patterns, including the Mediterranean diet, have demonstrated efficacy in enhancing the prognosis of NAFLD individuals, including their liver enzyme levels, liver fat, and histology [5,6]. Similarly, the healthy lifestyles of aerobic exercise, non-smoking, non-alcohol consumption, and adequate sleep have all been observed to have beneficial effects [7,8]. In addition, researchers found that a combination of aerobic exercise and intermittent fasting was effective in reducing liver steatosis in obese NAFLD patients in a randomized controlled trial [9]. Nevertheless, the evidence regarding the impact of healthy lifestyles, particularly sleep and psychological factors, on the reduction of the risk of all-cause and cardiovascular mortality in non-alcoholic fatty liver disease (NAFLD) is still limited. Furthermore, few studies have investigated the differences between subgroups of obese and non-obese NAFLD.

Currently, the treatment of NAFLD, like other metabolic diseases, is centered on lifestyle changes [4]. For example, high-quality diets (including the Mediterranean diet, intermittent diets, etc.) and aerobic exercise have been identified to improve the prognosis of NAFLD individuals [5,9].

Traditional lifestyles have been confirmed to have a relationship with lower all-cause mortality and cardiovascular disease mortality, such as avoidance of smoking, regular physical exercise, and healthy dietary [10]. Nevertheless, the specific influence of a healthy lifestyle, with particular emphasis on sleep quality and psychological health, on diminishing the mortality risk of NAFLD remains to be fully elucidated [11]. An observational study was conducted to evaluate the relationship between metabolic syndrome and sleep duration. The results demonstrated a U-shaped correlation, with individuals who slept for seven hours per night exhibiting the lowest likelihood of developing metabolic syndrome [12]. Mental status arises in the context of societal stressors, and the American Heart Association has recently emphasized the impact of mental health on cardiovascular health, pointing to the association between anxiety, depression, and pessimism and cardiovascular disease [13]. Psychological and sleep factors combined with traditional lifestyle and mortality outcomes in the NAFLD population deserve to be explored in depth, as this could provide more comprehensive and targeted interventions for patients with NAFLD. 

A retrospective matched cohort study was therefore performed, utilizing data from the National Health and Nutrition Examination Survey (NHANES) to explore the differences of healthy lifestyles on mitigating the risk of death in NAFLD individuals and normal populations. The aim was to identify which healthy lifestyle, or combination of healthy lifestyles, is more conducive to improving the prognosis of the NAFLD population.

## 2. Materials and Methods

### 2.1. Study Population

The data employed in this study are derived from NHANES, a nationally representative cross-sectional survey of the non-institutionalized U.S. resident population. [14]. The study sample consisted of seven consecutive cycles (2005–2018) of NHANES from which baseline participant information was collected. Data related to participant mortality were obtained through the National Center for Health Statistics (NCHS), and all-cause mortality data were obtained from the National Death Index as of October 2019. All data and materials were made publicly available on the NCHS website. The NHANES protocol was approved by the NCHS, Institutional Review Board of the U.S. Centers for Disease Control and Prevention, with the consent of all participants.

We excluded samples missing data on smoking, alcohol consumption, 24–h dietary review, exercise, and sleep. Samples missing covariates, pregnant women, and hepatitis B and C were not included in the study. To minimize reverse causality, we needed to exclude participants who died within 1 years before follow-up [15]. Thus, the first study sample included 5542 participants defined by the Hepatic Steatosis Index (HSI) and 5542 normal individuals matched to them according to age and poverty–income index (Appendix A). The second study sample included 1724 participants defined by the US Fatty Liver Index (USFLI) as a subsample used as a sensitivity analysis.

### 2.2. Definition of NAFLD

NHANES collected demographic data, physical examination data, laboratory data, dietary data, and lifestyle information from participants. NAFLD was defined using the HSI and USFLI in the absence of any causes of chronic liver disease and excessive alcohol consumption [16]. We calculated the HSI using this equation: HSI = 8 × (ALT/AST ratio) + body mass index (BMI) (+2, if diabetes; +2, if women). We defined NAFLD if HSI > 36. We used USFLI to screen out subsamples for use in sensitivity analyses. 

We calculated the USFLI using this equation: USFLI = (e^(0.3485^*^Mexican American − 0.8073^*^non-Hispanic black+0.0093^*^age+0.6151^*^InGGT+0.0249^*^waistcircumference+1.1792^*^ln(insulin)+0.8242^*^ln(glucose) − 14.7812)^/(1 + e^(0.3485^*^Mexican American − 0.8073^*^non-Hispanic black+0.0093^*^age+0.6151^*^InGGT+0.0249^*^waistcircumference+1.1792^*^ln(insulin)+0.8242^*^ln(glucose) − 14.7812)^)∗100. “Mexican American” and “non-Hispanic Black” have values of 1 if the participant is of that race and values of 0 if not. Subjects were defined as having NAFLD if they had a USFLI score of ≥30 and did not have (1) hepatitis B (hepatitis B surface antigen-positive) or hepatitis C infection (HCV-RNA-positive or hepatitis C antibody) or (2) excessive alcohol consumption (>1 alcoholic beverage/day for women or >2 alcoholic beverages/day for men).

Participants were defined as having obese NAFLD if their BMI was ≥30 kg/m^2^ and non-obese NAFLD if their BMI was <30 kg/m^2^ [17]. Participants were defined as having abdominal obese NAFLD if their waist circumference was ≥102 cm in men or ≥88 cm in women, and if their waist circumference was <102 cm in men or <88 cm in women were defined as non-abdominal obese NAFLD [18].

### 2.3. Definition of Lifestyle Factors

We referenced recent research presented at the American Society for Nutrition’s annual meeting, NUTRITION 2023, to incorporate mental health and sleep duration into healthy lifestyles [11]. Aggregate scores for healthy lifestyle factors were constructed by combining the number of healthy lifestyle elements, including smoking, physical activity, diet, sleep, and psychological factors. These data were obtained from a structured questionnaire and 24–h dietary recall. Then we categorized each lifestyle dichotomy, with non-smoking (smoking <100 cigarettes in whole life) defined as a healthy level, healthy physical activity level defined as ≥150 min of moderate physical activity or ≥75 min of vigorous physical activity during leisure time per week [19], the top 40% from high to low of the U.S. Healthy Eating Index (HEI–2015) considered to be a healthy diet [20], and healthy sleep defined as an average of 7–9 h of sleep per day [21]. The Patient Health Questionnaire (PHQ–9) is a nine-item depression screening instrument. It was administered to determine the frequency of depression symptoms over the past two weeks. A score of ≤4 on the PHQ–9 was used to define non-depressed psychological status [22].

We assigned a score of 0 to the unhealthy level and 1 to the healthy level of each lifestyle factor. The total score therefore ranges from 0 to 5, with higher values indicating healthier lifestyles [14]. Participants were categorized into three groups based on their lifestyle scores: those with an unfavorable lifestyle (0 and 1), an intermediate lifestyle (2 and 3), and a favorable lifestyle (4 and 5).

### 2.4. Follow-Up Time and Outcome

Follow-up time was from baseline to date of death or review, whichever came first, for all participants. Outcomes were all-cause mortality, cardiovascular mortality. Mortality outcomes were coded according to the International Classification of Diseases, 10th edition (ICD–10), for 2005–2018, including cardiovascular deaths (I00–I09, I11, I13, I20–I51, I60–I69). Premature mortality was defined as death before age 70 years [23].

### 2.5. Assessment of Covariates

The database was populated with a number of covariates, which were obtained through questionnaires and anthropometric assessments. These included age, gender (male or female), ethnicity (Mexican American, etc.), marital status (married/living with partner or widowed/divorced/separated), education level (less than high school, high school or higher), poverty status (poverty–income ratio), alcohol consumption (gram), history of cardiovascular disease (yes or no), and history of cancer (yes or no).

### 2.6. Statistical Analysis

All analyses were weighted to make the sample nationally representative, given NHANES’ complex sampling approach. T-statistics were used to analyze continuous variables and chi-square tests were used to compare characteristics of categorical variables with adjustments for sampling weights.

A one-to-one matching of 5542 NAFLD patients to 5542 normal individuals was conducted using a propensity score. Hazard ratios (HR) and 95% confidence intervals (CI) for mortality were calculated using survey-weighted multivariate Cox proportional risk regression models, which accounted for potential demographic and clinical confounders. Additionally, Kaplan–Meier curves were utilized for survival analysis. In Model 1, the following variables were adjusted for: gender, age, marital status, ethnicity, education, and poverty status. In Model 2, in addition to the aforementioned variables, BMI, cardiovascular disease, and cancer were also adjusted for.

For the purpose of subgroup analysis, the total population was divided into two subgroups, namely male and female. In the NAFLD population, the obese fatty liver subgroup was defined by body mass index (BMI ≥ 30), while the abdominal obese fatty liver subgroup was defined by waist circumference (≥102 cm for men and ≥88 cm for women). Age subgroups were divided using the median age of participants, 55 years, as the cutoff. We similarly performed the same analyses as before.

We calculated attributable risk ratios (AR%) to reflect the impact of NAFLD on mortality and evaluated the moderating effect of healthy lifestyle on AR%. AR% was calculated as follows: AR% = (HR − 1)/HR∗100. HR is the hazard ratio of NAFLD as an exposed [24,25].

All data were subjected to analysis using the R version 4.2.3 and SAS 9.4 software. A bilateral *p*-value of less than 0.05 (*p* < 0.05) was deemed to be statistically significant.

## 3. Results

### 3.1. Baseline Characteristics of the Study Cohort

Among the 11,084 eligible participants ultimately enrolled in the study, 5542 participants were identified with NAFLD according to the HSI, and an additional 5542 participants matched by age and poverty–income ratio were not diagnosed with NAFLD at the baseline level. The gender composition of the two populations differed significantly, and we therefore included gender subgroups in subsequent analyses (Table 1). Cases with NAFLD had higher BMI and WC values. For the five lifestyles, significant disparities were observed between the two groups, with participants free of NAFLD tending to be free of depression and exercising more.

### 3.2. Associations of Healthy Lifestyle with All-Cause and Cardiovascular Mortality in Individuals with/without NAFLD

After the median follow-up duration of 6.5 years for participants with NAFLD and 6.8 years for participants without NAFLD, there were totals of 502 and 541 deaths (Figure 1), respectively. Further Cox regression analysis showed that a healthy lifestyle decreased the risk of cardiovascular death in NAFLD participants compared to non-NAFLD individuals (HR, 0.54 [95% CI, 0.34–0.86] vs. HR, 0.56 [95% CI, 0.36–0.88] for sticking to an intermediate lifestyle; HR, 0.34 [95% CI, 0.20–0.59] vs. HR, 0.52 [95% CI, 0.27–1.00] for keeping to a favorable lifestyle). Each additional healthy lifestyle factor was independently associated with a significant reduction in the risk of both all-cause and cardiovascular disease mortality of NAFLD individuals (HR, 0.77 [95% CI, 0.67–0.88]; HR, 0.76 [95% CI, 0.63–0.91]), whereas the adoption of a healthy lifestyle only effectively decreased the risk of all-cause mortality in the population without NAFLD (HR, 0.76 [95% CI, 0.69–0.83]).

In a study investigating the impact of individual healthy lifestyle factors on the risk of all-cause and cardiovascular mortality, we observed that maintaining a non-depressed psychological status and adhering to vigorous physical activity were significantly associated with a reduced risk of cardiovascular mortality among NAFLD participants (HR, 0.64 [95% CI, 0.43–0.95]; HR, 0.54 [95% CI, 0.33–0.88]) compared to participants without NAFLD. Conversely adherence to vigorous physical activity and healthy sleep significantly reduced the risk of all-cause mortality in participants without NAFLD (HR, 0.59 [95% CI, 0.44–0.80]; HR, 0.73 [95% CI, 0.59–0.91]). In addition, the analysis of Model 2 verified the previous findings, and the results of the analysis were generally consistent with those of Model 1 (Appendix A). The AR% values of NAFLD for all-cause, cardiovascular, and premature death were 9.3% and 24.1% but were not statistically significant (Appendix A). Of these, vigorous exercise moderated 67.7% and 29.0% of the excess risk ratios of NAFLD for all-cause and cardiovascular death, respectively.

### 3.3. Associations of Healthy Lifestyle with All-Cause and Cardiovascular Mortality in Gender and Age Subgroups

In the analysis of gender subgroups, our findings revealed that conformity to aa beneficial lifestyle significantly decreased the risk of all-cause mortality across various subgroups (Figure 2 and Appendix A). Subsequently, we conducted an analysis to evaluate the impact of each individual healthy lifestyle factor on the risk of all-cause and cardiovascular mortality within distinct subgroups, in which, avoiding smoking significantly reduced the risk of all-cause mortality (Model 1: HR, 0.47 [95% CI, 0.34–0.66]; Model 2: HR, 0.50 [95% CI, 0.36–0.69]) in women with NAFLD, and maintaining a non-depressed psychological status, avoiding smoking, keeping to vigorous physical activity, and keeping a healthy diet significantly reduced their risk of cardiovascular mortality. However, these healthy lifestyles had a less effect than the former in reducing the risk of cardiovascular mortality for women without NAFLD. For men with NAFLD, maintaining a non-depressed mental status and healthy sleep significantly reduced the risk of all-cause mortality (HR, 0.55 [95% CI, 0.38–0.80]; HR, 0.64 [95% CI, 0.47–0.87]). It is noteworthy that maintaining a healthy diet was found to significantly reduce the risk of all-cause and cardiovascular mortality in male participants without NAFLD (HR, 0.69 [95% CI, 0.49–0.97]; HR, 0.54 [95% CI, 0.31–0.91]).

In the analysis of age subgroups, consistent engagement in beneficial healthy lifestyles significantly mitigated the risk of all-cause mortality among participants aged 55 years and older, irrespective of their NAFLD status. Maintaining a non-depressed psychological status and avoiding smoking significantly reduces all-cause mortality in NAFLD individuals aged 55 years and older (HR, 0.65 [95% CI, 0.47–0.92]; HR, 0.60 [95% CI, 0.46–0.79]) (Appendix A). Avoiding smoking and keeping a healthy diet significantly reduces all-cause mortality in participants with NAFLD under age 55 (HR, 0.40 [95% CI, 0.17–0.93]; HR, 0.36 [95% CI, 0.14–0.92]) (Appendix A).

### 3.4. Associations of Healthy Lifestyle with All-Cause and Cardiovascular Mortality in Participants with NAFLD and Obesity by BMI and WC

We used two indicators of physical examination, BMI, and WC, to define fat fatty liver subgroups-general obesity and abdominal obesity, respectively (Figure 3, Figure 4, Appendix A). The analysis showed that for both groups of participants, adherence to a favorable lifestyle helped reduce their risk of all-cause mortality (HR, 0.61 [95% CI, 0.44–0.85]; HR, 0.62 [95% CI, 0.47–0.83]), while each additional healthy lifestyle could significantly reduce the risk of cardiovascular death (HR, 0.70 [95% CI, 0.56–0.87]; HR, 0.76 [95% CI, 0.63–0.91]). Avoiding smoking reduced the risk of all-cause mortality in both groups (HR, 0.49 [95% CI, 0.36–0.66]; HR, 0.58 [95% CI, 0.43–0.77]). Survival analyses showed that adherence to vigorous physical activity reduced the risk of all-cause and cardiovascular mortality in both groups of participants. In addition, keeping a healthy sleep and diet did not significantly reduce the risk of all-cause and cardiovascular mortality in either group. Overall, the results for the two obese subgroups were generally consistent. It is of greater significance to note that our findings indicate that adherence to a healthy lifestyle does not appear to significantly reduce the risk of all-cause mortality in non-obese NAFLD patients, in contrast to the results observed in the two obese subgroups (Appendix A).

### 3.5. Correlations of Healthy Lifestyle with Premature Mortality in NAFLD Individual

We regarded deaths occurring before the age of 70 as premature deaths and examined whether a healthy lifestyle could influence the incidence of premature deaths. The analysis showed that adherence to a favorable lifestyle can significantly reduce premature deaths due to cardiovascular disease (HR, 0.73 [95% CI, 0.53–1.00]) (Figure 5). The evaluation of the influence of a single lifestyle on premature death showed that keeping to healthy sleep significantly reduced premature deaths due to cardiovascular disease (HR, 0.69 [95% CI, 0.50–0.95]). These findings from Model 2 are largely in concordance with the previous study. The AR% of NAFLD for premature death was 71.8% and significant. Of these, vigorous PA and healthy diet moderated 17.8% and 23.5% of the excess risk ratio of NAFLD for premature death, respectively (Appendix A).

### 3.6. Associations of Randomized Combinations of Healthy Lifestyle with All-Cause and Cardiovascular Mortality

Given that the above analyses suggest that adherence to a healthy lifestyle has a positive effect on improving prognosis in the NAFLD population, we identified specific clusters of lifestyle factors that were associated with the lowest mortality rates through comparative analyses of various combinations of healthy lifestyle factors against a null factor, considering only those combinations with an exposure of ≥5%. Multivariate analyses revealed a graded protective effect of a composite healthy lifestyle on the risk of both all-cause and cardiovascular mortality (Figure 6). For participants with NAFLD, maintaining a non-depressed mental status, avoiding smoking, and keeping a healthy sleep resulted in the lowest all-cause mortality. In both obese fatty liver subgroups, adherence to combinations of three or four lifestyles reduced all-cause and cardiovascular mortality. In contrast, among non-NAFLD participants, the risk of all-cause mortality was significantly reduced only when four or five healthy lifestyles were adhered to (Appendix A). More importantly, maintaining a non-depressed mental status occurred most frequently among the many combinations, suggesting that this lifestyle had the greatest impact on NAFLD prognosis.

### 3.7. Sensitivity Analyses Using Subsample by USFLI

We conducted sensitivity analyses by using the different hepatic steatosis index that define NAFLD (Appendix A). Overall, the results were predominantly congruent with those derived from the HSI. The subgroup with high lifestyle scores had a lower risk of death than the group with low scores, demonstrating a downward trend, albeit not reaching statistical significance. Each additional healthy lifestyle significantly reduced the risk of all-cause mortality in participants with NAFLD. In exploring the effect of a single healthy lifestyle on the risk of death, we also found a downward trend, but it was not significant.

## 4. Discussion

Our study, based on a large U.S. cohort, is the first to examine the impact of healthy lifestyles, including mood and sleep, on mortality in patients with NAFLD. For people with NAFLD, maintaining a healthy lifestyle is more effective in reducing cardiovascular mortality, which was our main finding. Notably, maintaining a non-depressed psychological status was more effective than other lifestyles in improving the prognosis of NAFLD patients. In the gender subgroup, healthy lifestyles improved the prognosis of female NAFLD patients better than that of male, while in the obese subgroup, the prognosis of both types of obese NAFLD patients improved significantly and the results were generally consistent. In addition, we note that maintaining a good lifestyle significantly reduces premature mortality due to cardiovascular disease in patients with NAFLD, in which keeping to healthy sleep plays an important role.

Numerous cohort studies have shown that traditional healthy lifestyles (including non-smoking, moderate exercise, and healthy diet) contribute to the reduction in mortality risk in both normal and chronically ill populations [10,26]. The incorporation of two additional lifestyles, namely mental state and sleep, into the analysis revealed that the addition of each healthy lifestyle reduced all-cause mortality by 23% and cardiovascular mortality by 24% in the NAFLD population. In contrast, the normal population exhibited a reduction in cardiovascular mortality of only 16%. Clearly, this is due to the high prevalence of cardiovascular outcomes in the NAFLD population. The average age of our included population was 53 years and, similar to the results of O’Doherty MG et al, maintaining a good healthy lifestyle in 50-year-old men and women prolongs life expectancy after the onset of cardiovascular disease [27].

We analyzed a single lifestyle to improve prognosis in a NAFLD population and found that psychological factors were more advantageous than traditional lifestyle in the male population. The risk of all-cause mortality was reported to be 1.32 times higher in the Chinese population (especially in men) with depression than in non-depressed patients [28]. Our study found that maintaining a positive psychological state reduced all-cause mortality in men by 45%, whereas avoiding smoking, vigorous exercise, and healthy sleep reduced it by 30%, 33%, and 36%, respectively. Moreover, in the analysis of multiple combinations of lifestyles, one and multiple combinations of lifestyles that included non-depressive states were significantly better than other combinations in reducing the risk of death. As an emerging lifestyle, the increased risk of cardiovascular death from inadequate sleep duration has been demonstrated in multiple cohorts [29,30], and maintaining a healthy sleep duration reduced the risk of all-cause mortality from NAFLD by 36%. In female patients, avoidance of smoking is particularly important to improving all-cause mortality outcomes, and reducing exposure to both firsthand and secondhand smoke significantly reduces nicotine harm. The NAFLD population is often comorbidly obese, with a minority of non-obese individuals, and previous studies have shown that these two groups have different metabolic disorders [31,32], with the former typically having insulin resistance and abnormal adipokine levels. Currently, BMI is commonly used in clinical practice as a surrogate measure of body fat and waist circumference as a measure of abdominal obesity [18]. We explored the difference between healthy lifestyles on the prognosis of fat and obese patients by dividing the group by two obesity types. Interestingly, none of the five healthy lifestyles significantly reduced all-cause mortality in the non-obese NAFLD, whereas the two obese types had essentially the same results, with benefits in all-cause mortality and cardiovascular mortality. Therefore, waist circumference could also be used as an indicator to independently classify the degree of obesity in the NAFLD population. A cohort study based on a U.S. population with up to 34 years of follow-up found that women with five low-risk factors had an average life expectancy at age 50 that was 14.0 years longer (95% CI, 11.8–16.2) compared to women without the low-risk factors, while the difference for men was 12.2 years (95% CI, 10.1–14.2) years [33]. Not only that but adopting a healthy lifestyle can also play a role in reducing premature mortality and extending the life expectancy of people with NAFLD.

These five important healthy lifestyles are closely related to the body’s metabolism, immunity, biorhythms, and other mechanisms. Chronic smoking induces oxidative metabolism of glucose and inhibits non-oxidative reactions, ultimately leading to elevated plasma free fatty acid (FFA) levels, which accelerates the progression of NAFLD [34]. Adverse factors in dietary patterns, especially red meat, and high cholesterol intake, may increase oxidative stress in the liver and promote progression of NAFL to NASH [35]. An unhealthy diet can be balanced by exercise. Resistance exercise may change muscle characteristics and improve hepatic steatosis by upregulating glycolysis and improving insulin resistance [7]. Inadequate sleep duration and poor sleep quality can affect metabolic function in people with type 2 diabetes by causing insulin resistance, increased appetite and impaired glucose tolerance [36]. More interestingly, decreased sleep duration also increases daytime fatigue, which may lead to the poor outcome of depression [37]. It is becoming increasingly evident that chronic inflammation and increased oxidative stress may play a role in the progression of depression and NAFLD/MAFLD. [38,39]. Increased monoamine oxidase-A (MAO-A) activity was found in patients with depression and NAFLD [40,41], which may be associated with enhanced cellular oxidative stress and adversely affect liver pathology in NAFLD [41].

This study utilized seven cycles of NHANES population data from 2005–2018, which is a sufficient sample size for the findings to be generalized to a more extensive U.S. population. More importantly, our study can provide a scientific basis for healthy lifestyles to improve the prognosis of NAFLD in patient populations, including men and women, obese and non-obese NAFLD populations. A sensitivity analysis was conducted to cross-validate the reliability of the findings by utilizing two indices to diagnose fatty liver. It should be noted that the present study is not without limitations. On the one hand, the lifestyle recall data was obtained via a questionnaire, which introduces a number of confounding factors into the study. On the other hand, our study is limited to the U.S. population and therefore cannot be generalized to other countries or other ethnic groups. Because of the lack of follow-up data in the NAHNES database, we only considered participants with NAFLD at baseline and ignored those who developed NAFLD during follow-up because of progression of fatty liver. Finally, healthy lifestyles reduced all-cause mortality in non-obese NAFLD participants, although the findings did not achieve statistical significance. Consequently, further investigation through large-scale prospective cohort studies is warranted to delineate optimal lifestyle interventions for NAFLD.

## 5. Conclusions

Patients with NAFLD are in greater need of emerging healthy lifestyles combinations to reduce mortality compared to the normal population, especially women and the obese.

## Figures and Tables

**Figure 1 nutrients-16-02063-f001:**
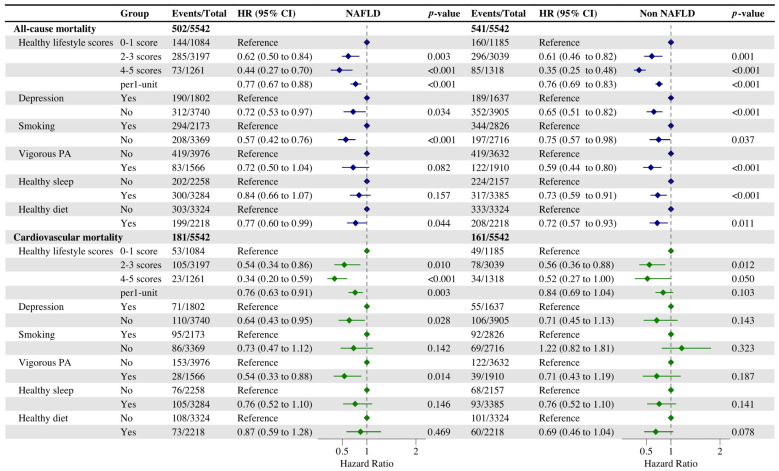
Cox regression analysis of the association between healthy lifestyle and all-cause and cardiovascular mortality. Adjusted for gender, age, marital status, ethnicity, education, and poverty status. *p* < 0.05 was considered significant.

**Figure 2 nutrients-16-02063-f002:**
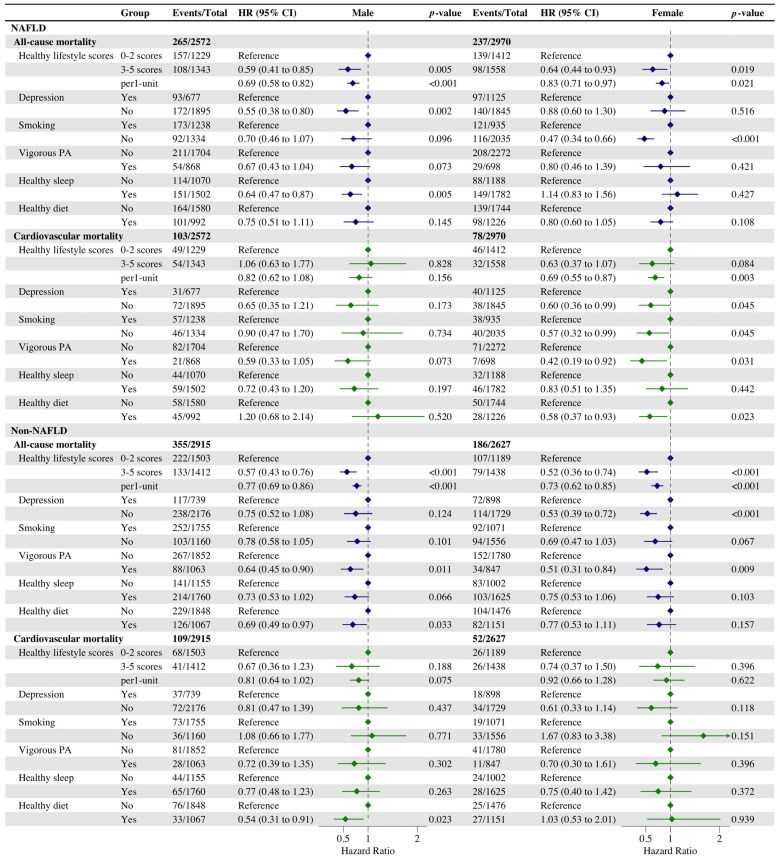
Cox regression analysis of the association between healthy lifestyle and all-cause and cardiovascular mortality in men and women with NAFLD. Adjusted for gender, age, marital status, ethnicity, education, and poverty status. *p* < 0.05 was considered significant.

**Figure 3 nutrients-16-02063-f003:**
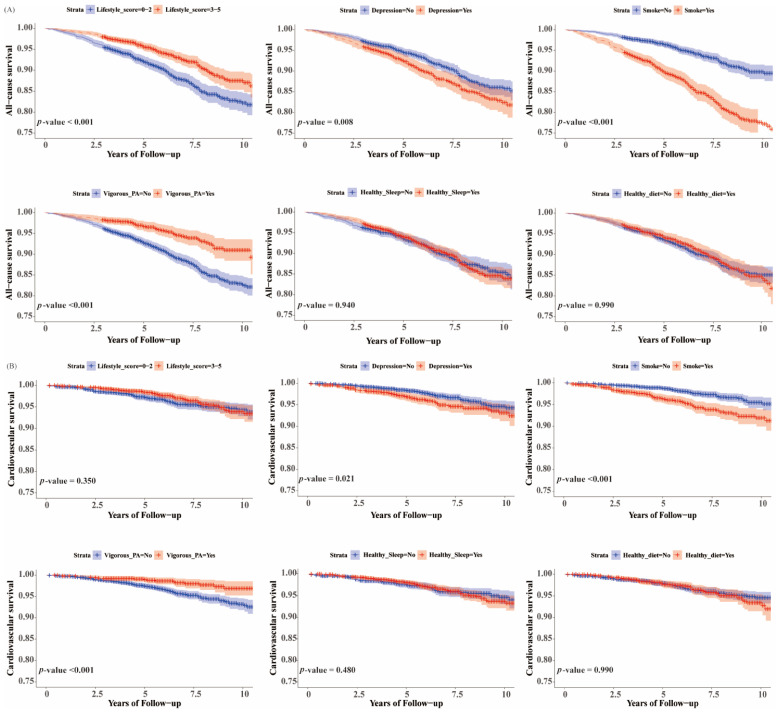
Kaplan–Meier survival curve for all-cause and cardiovascular mortality stratified on lifestyle category. (**A**) All-cause mortality for NAFLD by BMI; (**B**) cardiovascular mortality for NAFLD by BMI.

**Figure 4 nutrients-16-02063-f004:**
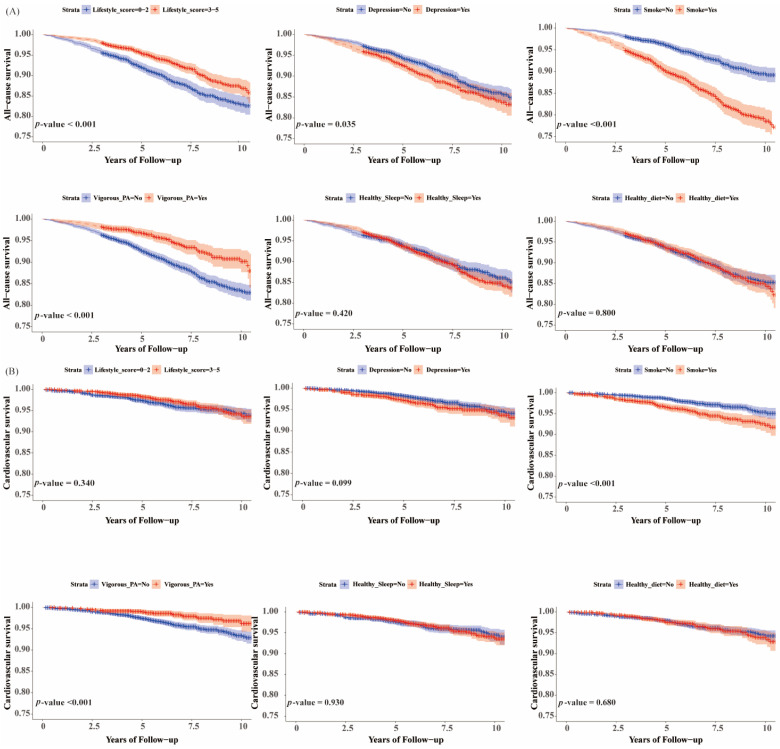
Kaplan–Meier survival curve for all-cause and cardiovascular mortality stratified on lifestyle category. (**A**) All-cause mortality for NAFLD by waist circumference; (**B**) cardiovascular mortality for NAFLD by waist circumference. *p* < 0.05 was considered significant.

**Figure 5 nutrients-16-02063-f005:**
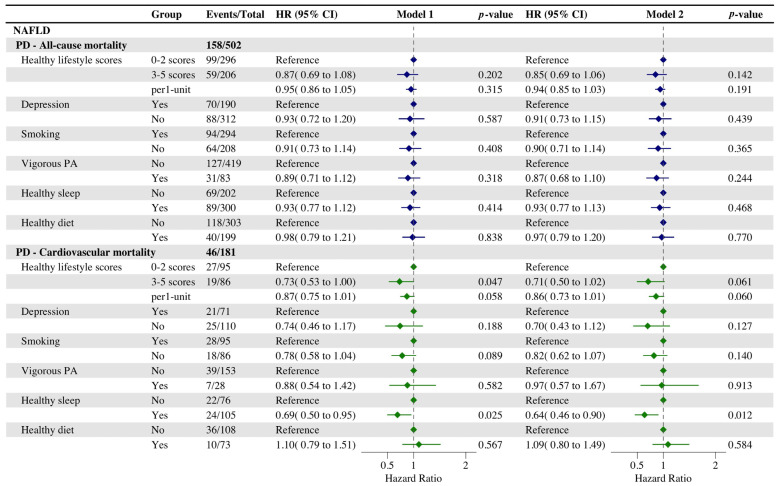
Cox regression analysis of the association between healthy lifestyles and premature mortality in participants with NAFLD. Model 1 was adjusted for gender, age, marital status, ethnicity, education, and poverty status. Model 2 was adjusted for gender, age, marital status, ethnicity, education, and poverty status, BMI, cardiovascular disease, and cancer. *p* < 0.05 was considered significant. Abbreviation: PD, premature death.

**Figure 6 nutrients-16-02063-f006:**
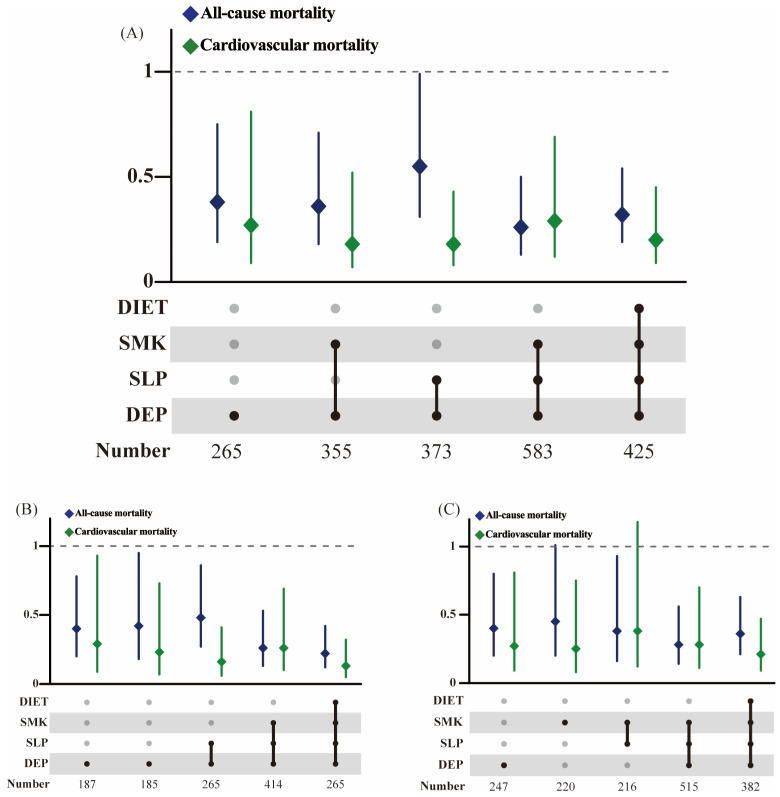
Association between different combinations of healthy lifestyles and all-cause and cardiovascular mortality in NAFLD participants. Adjusted for gender, age, marital status, ethnicity, education, and poverty status. Abbreviation: SMK, smoking; SLP, sleeping; DEP, depression. (**A**) For participants from 5542 participants defined by the HSI; (**B**) for participants with NAFLD and obesity by BMI; (**C**) for participants with NAFLD and obesity by waist circumference.

**Table 1 nutrients-16-02063-t001:** Characteristics of the study population (*n* = 11,084).

Characteristics	Non-NAFLD Population (N = 5542)	NAFLD Population (N = 5542)	*p*-Value
Demographic information			
Age, years, mean (SE)	53.5 (0.2)	53.4 (0.2)	0.904
Gender, *n* (%)			<0.001
Male	2915 (52.6)	2572 (46.4)	
Female	2627 (47.4)	2970 (53.6)	
Ethnicity, *n* (%)			<0.001
Mexican American	770 (13.9)	902 (16.3)	
Other Hispanic	540 (9.7)	595 (10.7)	
Non-Hispanic White	2578 (46.5)	2343 (42.3)	
Non-Hispanic Black	1019 (18.4)	1227 (22.1)	
Other race, including multiracial	635 (11.5)	475 (8.6)	
Marital status, *n* (%)			0.010
Married/Living with partner	3377 (60.9)	3509 (63.3)	
Widowed/Divorced/Separated	2165 (39.1)	2033 (36.7)	
Educational level, *n* (%)			0.085
Less than high school	1269 (22.9)	1346 (24.3)	
High school or higher	4273 (77.1)	4196 (75.7)	
Poverty–income ratio, mean (SE)	2.5 (0.02)	2.5 (0.02)	0.469
Physical examination and laboratory analysis			
Body mass index, kg/m², mean (SE)	26.8 (0.1)	33.0 (0.1)	<0.001
Waist circumference, cm, mean (SE)	95.0 (0.2)	108.8 (0.2)	<0.001
ALT, IU/L, mean (SE)	23.5 (0.2)	27.5 (0.2)	<0.001
AST, IU/L, mean (SE)	25.9 (0.2)	25.3 (0.1)	0.034
Lifestyle and diseases			
No depression, *n* (%)	3905 (74.5)	3740 (67.5)	<0.001
No smoking, *n* (%)	2716 (49.0)	3369 (60.8)	<0.001
Vigorous PA, *n* (%)	1910 (34.5)	1566 (28.3)	<0.001
Healthy sleep, *n* (%)	3385 (61.1)	3284 (59.3)	0.050
HEI–2015 score, mean (SE)	52.5 (0.2)	50.8 (0.2)	<0.001
Alcohol assumption, gram, mean (SE)	34.6 (0.5)	9.0 (0.1)	<0.001
CVD, *n* (%)			<0.001
No	4950 (89.3)	4803 (86.7)	
Yes	592 (10.7)	739 (13.3)	
Cancer, *n* (%)			0.436
No	46,902 (88.5)	4928 (88.9)	
Yes	640 (11.6)	614 (11.1)	

**Note:** Descriptive data were shown as mean (SE), while categorical variables were reported as n (%). *p* < 0.05 was considered significant. Abbreviation: NAFLD, nonalcoholic fatty liver disease; ALT, alanine aminotransferase; AST, aspartate transaminase; PA, physical activity; HEI, healthy eating index; CVD, cardiovascular disease; SE, standard error; *n*, number.

## Data Availability

The National Health and Nutrition Examination Survey 2005–2018 datasets are publicly available at the National Center for Health Statistics of the Center for Disease Control and Prevention (https://www.cdc.gov/nchs/nhanes/index.htm, access on 5 November 2023).

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
