# Peer review of "Association between Lifestyle Modification and All-Cause, Cardiovascular, and Premature Mortality in Individuals with Non-Alcoholic Fatty Liver Disease"

_nutrients, 2024, doi:10.3390/nu16132063_

Round 1
Reviewer 1 Report
Comments and Suggestions for Authors
This is a great reseacrch that addresses and important concern especially given the growing epidemic of obesity and its associated complications like NAFLD now known as MASLD (metabolic dysfunction associated steatotic liver disease.
Comments:
Is there a reason for using the old terminology (NAFLD) and not MASLD?
Section 2.4 talked about follow-up from the time the participant was diagnosed. How did you determine the time of diagnosis? There may be a need to re-phrase this. Sounds like the follow-up data was from the time of initial data collection ( which was when you calculated the HSI.
Also you categorized the lifestyle measures as emerging. I would refrain from calling Healthy diet, avoidance of smoking and physical activity as emerging lifestyle. They have been studied extensively. However, the novel thing about your study is using multiple healthy lifestyles combination which is a brilliant idea.
One other limitation of the study, is that the some of non NAFLD population could also have developed NAFLD by the time of follow-up and that could have affected the result for that group.
Author Response
Comment 1:
Is there a reason for using the old terminology (NAFLD) and not MASLD?
Response 1: Thank you for the comment. In our study, hepatic steatosis was diagnosed by calculating the HSI and USFLI based on the metabolic indicators of the participants. However, the diagnostic criteria for MAFLD need to fulfil metabolic dysregulation in addition to hepatic steatosis. This duplicates the metabolic indicators required for the previous HSI and USFLI. Therefore, to avoid bias, we considered it more appropriate to diagnose NAFLD.
Comment 2:
Section 2.4 talked about follow-up from the time the participant was diagnosed. How did you determine the time of diagnosis? There may be a need to re-phrase this. Sounds like the follow-up data was from the time of initial data collection (which was when you calculated the HSI.
Response 2: Thanks for your suggestion and we have adapted it in the manuscript (page 4 of 19, lines 155-156). We calculated HSI using relevant indicators at baseline for all participants, regardless of whether they were diagnosed with NAFLD or not, with follow-up from baseline to time of death or review, whichever came first.
Comment 3:
Also you categorized the lifestyle measures as emerging. I would refrain from calling Healthy diet, avoidance of smoking and physical activity as emerging lifestyle. They have been studied extensively. However, the novel thing about your study is using multiple healthy lifestyles combination which is a brilliant idea.
Response 3: Thank you for your suggestion and we have adapted it in the manuscript.
Comment 4:
One other limitation of the study, is that the some of non NAFLD population could also have developed NAFLD by the time of follow-up and that could have affected the result for that group.
Response 4: Thank you for your suggestion and we have adapted it in the manuscript (page 15 of 19, lines 443-445). We do agree with your opinion that the some of non NAFLD population could also have developed NAFLD by the time of follow-up and that could have affected the result for that group. However, NHANES only has information on baseline and date of death or review. Thus, it is difficult to evaluate and exclude whether non-NAFLD individuals progressed to NAFLD during the follow-up period,. Accordingly, we further added this limitation in the revised manuscript.
Reviewer 2 Report
Comments and Suggestions for Authors
I have studied the manuscript entitled "Association between Lifestyle Medicine and All-cause, Cardiovascular and Premature Mortality in Individuals with Non-Alcoholic Fatty Liver Disease" by Huang Y.
It is rather unclear to what extent the referred lifestyle modifications could benefit the NAFLD patients group in comparison with the control group. The authors are kindly encouraged to explicitly provide the excess risk in terms of all-cause mortality, cardiovascular mortality, and premature mortality attributable to NAFLD (now MAFLD); for this purpose, see relevant papers (e.g. Chen YQ, Hu C, Wang Y. Attributable risk function in the proportional hazards model for censored time-to-event. Biostatistics. 2006;7:515-29. PMID: 16478758). Moreover, they are encouraged to explain how this excess risk is affected by lifestyle modifications. This approach, if given clearly and comprehensively, would ameliorate the manuscript and consolidate the conclusions.
Comments on the Quality of English LanguageMinor editing of English language required.
Author Response
Comment 1:
It is rather unclear to what extent the referred lifestyle modifications could benefit the NAFLD patients group in comparison with the control group. The authors are kindly encouraged to explicitly provide the excess risk in terms of all-cause mortality, cardiovascular mortality, and premature mortality attributable to NAFLD (now MAFLD); for this purpose, see relevant papers (e.g. Chen YQ, Hu C, Wang Y. Attributable risk function in the proportional hazards model for censored time-to-event. Biostatistics. 2006;7:515-29. PMID: 16478758). Moreover, they are encouraged to explain how this excess risk is affected by lifestyle modifications. This approach, if given clearly and comprehensively, would ameliorate the manuscript and consolidate the conclusions.
Response 1: Thank you for your suggestion and we have adapted it in the manuscript (page 5 of 19, lines 187-190; page 6 of 19, lines 232-235; page 12 of 19, lines 311-313). We added Table S1 to assess the attributable risk (AR) of NAFLD for mortality and the moderating effect of healthy lifestyle on the attributable risk of NAFLD. AR% is calculated as follows:AR% = (HR-1)/ HR *100. HR is the hazard ratio of NAFLD as an exposed. We found that the AR% of NAFLD for all-cause, cardiovascular, and premature mortality were 9.3%, 24.1%, and 71.8%, respectively, and were statistically significant only for premature death (Table S1).
Round 2
Reviewer 2 Report
Comments and Suggestions for Authors
I have carefully studied the revised version of the manuscript entitled "Association between Lifestyle Medicine and All-cause, Cardiovascular and Premature Mortality in Individuals with Non-Alcoholic Fatty Liver Disease".
The authors have successfully responded to the queries raised during the review process. The quality of the manuscript has been improved. There are no additional comments/issues.
Comments on the Quality of English LanguageMinor editing of English language required.